# Explainable Forensics of Manipulated Segments in Untrimmed Long Videos

Yue Feng [1] [*]   Jingjing Li [2] [*]   Qijia Lu [1] [*]   Wei Ji [3] [‡] [†]   Jingrou Zhang [1]   Fei Shen [4]   Xiao Li [1]   Yizhen Jia [1]
Qiang Chen [3] [‡]   Limin Wang [5]   Wentong Li [1] [†]   Jie Qin [1] [†]

## Abstract

The rapid advancement of AI-driven video generation has transformed content creation, while simultaneously increasing the risk of misinformation through localized manipulations in long-form videos. Existing video forensic methods predominantly operate on short, independent clips, and thus fail to capture realistic scenarios where AI-generated content is sparsely embedded within otherwise authentic footage. To bridge this gap, we formulate the task of *Temporal AI-Generated Segment Localization and Explanation*, which targets authenticity detection, temporal localization, and interpretable analysis of manipulated segments in untrimmed long videos. We further introduce TASLE, a large-scale benchmark comprising 12,472 untrimmed videos with diverse manipulation patterns and rich annotation signals, including temporal boundaries, authenticity labels, and segment-level rationales. In addition, we propose MSLoc, a coarse-to-fine forensic baseline that combines a boundary-sensitive proposal generation module for efficient long-video scanning with an MLLM-based refinement module for precise boundary localization and interpretable reasoning. Experiments validate the effectiveness of the proposed baseline, highlighting the importance of segment-level explainable forensics for long-form AI-generated video analysis. Our dataset and code are publicly available at https://debby-0527.github.io/TASLE.

[*]Equal Contribution [†]Corresponding Author [‡]Project Lead
[1]MoE Key Laboratory of Brain-Machine Intelligence Technology, College of Artificial Intelligence, Nanjing University of Aeronautics and Astronautics [2]Dalian University of Technology [3]Independent Scholar (ethan.chen1988@gmail.com) [4]National University of Singapore [5]Nanjing University. Correspondence to: Wei Ji <weiji.yale@gmail.com>, Wentong Li <wentong_li@nuaa.edu.cn>, Jie Qin <jie.qin@nuaa.edu.cn>.

*Proceedings of the 43$^{rd}$ International Conference on Machine Learning*, Seoul, South Korea. PMLR 306, 2026. Copyright 2026 by the author(s).

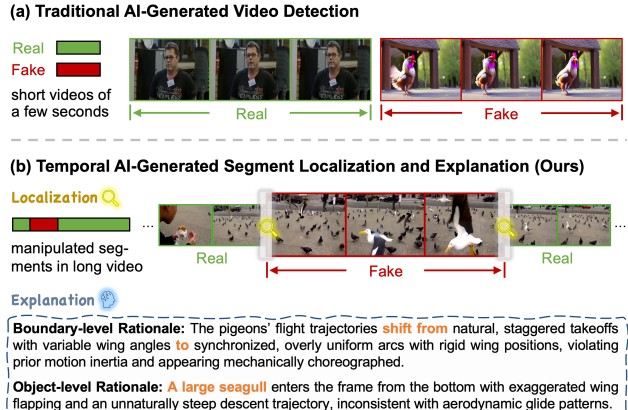

*Figure 1.* **Illustration of the traditional AI-generated video detection paradigm and the proposed long-video AI-generated segment localization and explanation task**. (a) Traditional methods operate on short video clips and perform binary real–fake classification, without modeling mixed real–fake contexts in long videos. (b) In contrast, our task considers long-form videos with sparsely embedded AI-generated segments, aiming to localize manipulated intervals and provide interpretable forensic rationales at both boundary and object levels.

## 1. Introduction

Recent advances in AI-driven video generation have significantly reshaped content creation, enabling high-quality video synthesis with increasing realism and controllability. Creators can generate complex visual narratives directly from natural language descriptions (Team et al., 2025), optionally conditioned on reference images or visual instructions (*e.g.*, masks) (Wan et al., 2025; Jiang et al., 2025b). As a result, the barrier to professional-level video production has been substantially lowered, driving adoption across application domains such as film production, advertising, and digital media (Zhang et al., 2025b; Hashim et al., 2025; Xu et al., 2026; Li et al., 2026; Feng et al., 2026a).

It is worth noting that the same properties that empower creativity also lead to new challenges for content authenticity (Golda et al., 2024). AI-generated video techniques can be intentionally misused to falsify events, manipulate object behaviors, or alter subject actions in real scenarios. For example, altering the semantics of road signs in video footage may cause autonomous driving systems to misinterpret criti-

*Table 1.* **Overview of TASLE and existing AI-generated video detection datasets**. "Entire" and "Mixed" indicate whether a dataset contains fully generated videos or temporally mixed (partially manipulated) videos. "Det.", "Loc.", and "Exp." denote support for detection, temporal localization, and explanation tasks, respectively. T2V: Text-to-Video; I2V: Image-to-Video; TV2V: Text-Video-to-Video; FLF2V: First-Last-Frame-to-Video; TI2V: Text-Image-to-Video; MV2V: Mask-Video-to-Video.

| Dataset | Video Num. | Video Dura. | AIGC Dura. | Entire | Mixed | Det. | Loc. | Exp. | AIGC Types |
|---|---|---|---|---|---|---|---|---|---|
| GenVideo (Chen et al., 2024) | 2,302k | 2-6s | 2-6s | ✓ | ✗ | ✓ | ✗ | ✗ | T2V, I2V |
| GenVidBench (Ni et al., 2026) | 144k | 1-2s | 1-2s | ✓ | ✗ | ✓ | ✗ | ✗ | T2V, I2V |
| GenBuster (Wen et al., 2025) | 200k | 5s | 5s | ✓ | ✗ | ✓ | ✗ | ✗ | T2V |
| GenWorld (Chen et al., 2025) | 100k | 1-20s | 1-5s | ✓ | ✗ | ✓ | ✗ | ✗ | T2V, I2V, TV2V |
| TASLE | 12.5k | 2-124s | 1-15s | ✓ | ✓ | ✓ | ✓ | ✓ | FLF2V, TI2V, MV2V |

cal traffic instructions, potentially leading to unsafe control decisions (Chi et al., 2025; Ke et al., 2025; Feng et al., 2026b). Such targeted manipulations enable the injection of misinformation into videos, posing significant risks to the reliability and trustworthiness of visual media (Xu et al., 2024; Jia et al., 2025; Yan et al., 2025). These concerns have driven growing interest in AI-generated video detection. However, current methods such as DeMamba (Chen et al., 2024) and BusterX (Wen et al., 2025) are primarily designed for short video clips, as illustrated in Fig. 1(a). They focus on discriminating clips with durations of only a few seconds, typically assuming that each clip is independently either real or fake. This formulation overlooks a more realistic scenario in which AI-generated content appears only in localized segments of a long-form video, while the remaining portions remain authentic.

The lack of mechanisms to model mixed AI–real contexts and long-range temporal dependencies fundamentally limits the effectiveness of these methods. As shown in Fig. 1(b), real-world cases often involve subtle, sparse, and temporally localized manipulations embedded within coherent real-video contexts, making detection substantially more challenging than short-clip settings. Meanwhile, as listed in Table 1, existing benchmark datasets (*e.g.*, GenVidBench (Ni et al., 2026)), which serve as critical infrastructure for model training and evaluation, remain largely limited to isolated real short videos and fully synthetic fake clips, and thus fail to reflect the complexity of long-form scenarios (Xu et al., 2025b).

Motivated by these observations, we study the problem of forensic detection of manipulated segments in untrimmed long videos, an important yet under-explored setting where AI-generated content (AIGC) is embedded within otherwise authentic footage. As shown in Fig. 1(b), we refer to this problem as *Temporal AI-Generated Segment Localization and Explanation*, a task formulation of explainable forensics in long videos. To facilitate systematic research, we introduce a dedicated benchmark built upon a large-scale long-video dataset, termed **TASLE**. The dataset features diverse manipulation patterns, including temporally coherent AIGC content conditioned on segment-level reference frames or instructions, as well as localized object-level edits. It comprises 12,472 long untrimmed videos ranging from 2 to 124 seconds, covering diverse sources and challenging scenes. TASLE also provides rich annotation signals, consisting of precise temporal localization, authenticity classification labels, and segment-level rationales that summarize boundary- and object-level cues in context. These elements enable systematic analysis of long-context forensic reasoning and support principled evaluation of both localization and interpretability.

Furthermore, we present a baseline framework, **MSLoc**, tailored to the long-video forensics setting. Rather than treating long videos as collections of independent short clips, MSLoc explicitly leverages the sparse and boundary-centric nature of real-world manipulations. It adopts a coarse-to-fine design that decomposes the task into efficient proposal filtering and boundary-aware refinement. In the first stage, MSLoc performs lightweight proposal generation by scanning long videos with a boundary-sensitive learning objective, enabling the model to focus on temporal transition regions where real and manipulated content intersect. This design improves localization recall while avoiding exhaustive processing of authentic sequences. In the second stage, MSLoc refines the shortlisted proposals using a localization-oriented MLLM with a cue-driven anomaly-aware learning objective, which jointly enables precise temporal localization and interpretable forensic rationales.

Extensive experiments validate the effectiveness and rationality of the proposed baseline, and showcase the significance of forensic detection of manipulated segments in untrimmed long videos. In summary, this work offers a unified benchmark with long-video data, rich explainability annotations, and a dedicated baseline, enabling principled evaluation and advancing long-form AI-generated video forensics. We will release the dataset and code to facilitate reproducibility and encourage future research on this task.

## 2. Related Work

**AI-Generated Video Detection.** Early approaches for detecting AI-generated videos primarily relied on deep

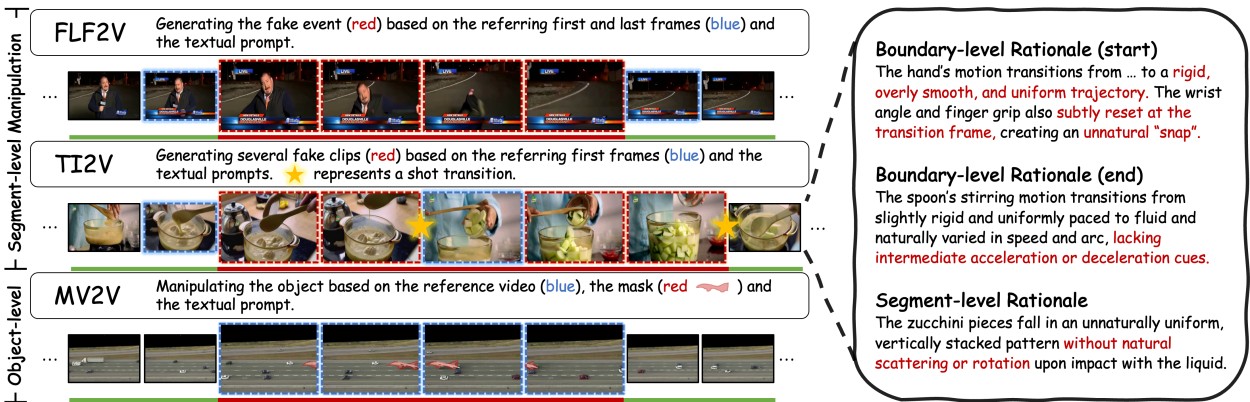

*Figure 2.* **Overview of the manipulation patterns and annotation granularity in TASLE**. We employ three advanced generative paradigms: *FLF2V* and *TI2V* for segment-level generation, and *MV2V* for object-level manipulation. Green borders indicate real reference frames or videos, while red borders highlight the AI-generated segments or masks. The right panel exemplifies the annotated rationales, comprising *boundary-level rationales* that describe inconsistencies at the transition regions (*e.g.*, unnatural motion state) and *object-level rationales* that explain visual anomalies within the manipulated content.

learning backbones to extract spatial-temporal artifacts. These methods often utilize Mamba or Transformer architecture (Li et al., 2025b) to capture pixel-level inconsistencies or frequency domain anomalies (Chen et al., 2024). Subsequent work incorporates physical priors, such as D3 (Zheng et al., 2025) which introduces a training-free method based on second-order motion features derived from Newtonian mechanics. With the advent of Multimodal Large Language Models (MLLMs), the focus has shifted towards explainable detection. FakeShield (Xu et al., 2025c) and IVY-FAKE (Jiang et al., 2025a) employ MLLMs to provide natural language explanations for forgeries. More recently, DAVID-XR1 (Gao et al., 2025) and AvatarShield (Xu et al., 2025d) explore fine-grained defect analysis and reinforcement learning to enhance forensic reasoning. However, existing methods and benchmarks (e.g., GenVideo (Chen et al., 2024), GenVidBench (Ni et al., 2026)) predominantly focus on binary classification of short clips, lacking mechanisms to handle sparse, localized manipulations within long-form videos. To address this gap, we introduce the TASLE benchmark specifically designed for the localization of sparse manipulated segments in long videos.

**MLLM-based Temporal Video Grounding.** Temporal video grounding aims to localize specific moments in a video based on textual queries (Feng et al., 2023; Ma et al., 2025). The emergence of video-based LLMs has significantly advanced this field. TimeChat (Ren et al., 2023) pioneers time-sensitive video understanding via a timestamp-aware frame encoder and a sliding video Q-Former. To improve localization precision and reasoning, VTimeCoT (Zhang et al., 2025a) proposes a visual chain-of-thought framework that explicitly reasons about temporal boundaries using a progress bar tool, while Trace (Guo et al., 2024) formulates video outputs as causal events and employs task-interleaved generation for timestamps, scores,

and captions. Although effective for semantic event localization, applying these methods directly to forgery detection is challenging due to the subtle nature of generative artifacts and the high computational cost for long videos (Feng et al., 2026c). Our MSLoc model addresses this by adopting a coarse-to-fine localization strategy that balances processing efficiency with sensitivity to subtle manipulation artifacts in long-form videos.

## 3. TASLE Dataset

In this section, we describe the construction of the TASLE dataset and present key statistical characteristics.

**Data Collection.** Our goal is to collect a large-scale data corpus that reflects realistic long-video scenarios encountered in the real world. To this end, we collect long-form videos from 11 diverse categories across multiple public sources, including sports, egocentric interactions, entertainment, and other open-world settings. These videos exhibit complex scenes, long temporal structures, and rich semantic content, which are critical for modeling realistic mixed AI–real contexts. See Appx. A for sourced data details.

**Data Processing and Annotation.** To realistically simulate the presence of AI-generated content in long-form videos, we design a controlled segment creation and insertion protocol that preserves temporal coherence and semantic consistency with the surrounding real-video context. Specifically, we adopt a human-in-the-loop pipeline (Fig. 7) that models two levels of manipulation: *segment-level* and *object-level* generation (Fig. 2). (i) *Segment-level generation.* Build upon temporal localization annotations from the source datasets, we can firstly identify candidate video segments corresponding to salient actions or events. Then large vision-language model is employed to falsify event seman-

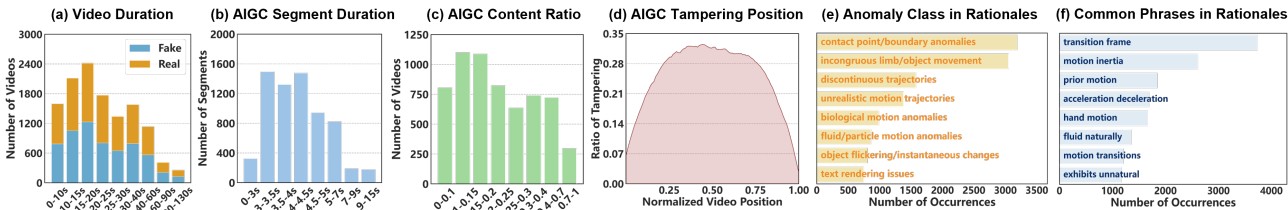

*Figure 3.* **Statistics of our TASLE dataset** in terms of (a) video duration, (b) AIGC segment duration, (c) AIGC content ratio, (d) AIGC tampering position, as well as anomaly class (d) and common phrases (d) in rationales.

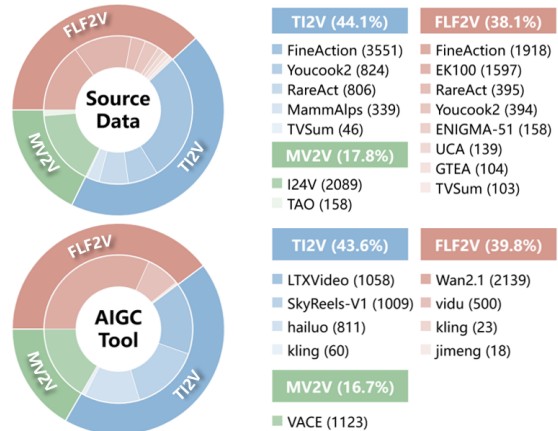

*Figure 4.* **Distribution of source data and AIGC tools used in the proposed TASLE Dataset**. More details are in Tables 8 and 9.

tics. To avoid overfitting to the original segment distribution of the source datasets, we perform random segment selection and temporal clipping. Based on the resulting textual descriptions and reference frames, we produce AI-generated video segments with various FLF2V and TI2V generators, *e.g.*, (Wan et al., 2025; Bao et al., 2024). For generators without explicit control over ending frames, shot detection is applied to ensure temporal coherence. (ii) *Object-level generation.* We leverage spatio-temporal annotations to falsify salient objects (*e.g.*, pedestrians, interaction targets, and vehicles). Concretely, we apply SAM2-based (Ravi et al., 2024) video object segmentation to obtain object masks, discarding objects that are too small or persist for excessively long durations. Targets are then replaced or removed using a mask-conditioned video-to-video (MV2V) generation tool (Jiang et al., 2025b), producing fine-grained object-level manipulations. Fig. 2 presents visual examples.

During processing, all videos are standardized to a resolution of $832 \times 480$ at 15 FPS to ensure compatibility with common AI video generation tools. The generated segments are seamlessly integrated back into the original long videos, followed by automatic consistency checks on resolution, frame count, and temporal alignment. Finally, human inspection by six annotators is conducted to verify visual

quality and transition smoothness, resulting in realistic and challenging manipulation cases.

**Rationale Annotation.** To support explainable forensics, we aim to provide concise and faithful rationales that highlight visually grounded AIGC cues within manipulated segments. A key challenge lies in the high realism of our AI-generated content: since segments are created under strong reference-frame constraints, both static appearance and motion dynamics remain coherent, making purely generative explanations prone to hallucination. To address this issue, we adopt a *comparative annotation strategy*. Specifically, for each AI-generated segment, we provide the annotation model with a corresponding reference segment extracted from the original real video (or a masked video in the case of object-level generation). For object-level generation, the annotation model additionally receives the manipulation mask to precisely identify which objects have been altered, and structured prompts explicitly instruct the model to describe anomalies in the masked regions. This design encourages explicit comparison between real and AI-generated content, guiding the model to focus on discriminative visual differences rather than speculative artifacts. To further facilitate fine-grained spatio-temporal alignment, we vertically concatenate the reference and AI-generated videos, enabling direct frame-wise comparison. In addition to *segment-level rationales*, we introduce *boundary rationales* to explain transitions between real and AI-generated content in long videos, as illustrated in Fig. 2. These boundary annotations capture temporal cues around manipulation onsets and offsets, supporting interpretable boundary localization. Using carefully designed prompts, we apply a large multimodal video-language model (Qwen3-VL-235B (Bai et al., 2025)) to generate both segment-level and boundary-level rationales. Finally, all rationales undergo manual screening and correction by six inspectors to ensure salience, accuracy, and consistency.

**Statistical Analysis.** Here we present a statistical analysis of the proposed TASLE dataset across several aspects. *(1) Distribution.* Table 1 summarizes key characteristics of TASLE in comparison with existing AI-generated video forensics datasets. TASLE contains 12,472 untrimmed videos with durations ranging from 2 to 124 seconds, sub-

stantially extending beyond prior datasets that focus on short clips. Fig. 3(a) and (b) present the distribution of video duration and AIGC segment. Fig. 3(c) reports the ratio of manipulated segments relative to entire videos, showing that AI-generated content typically occupies a small portion of the video (mostly ≤30%). Fig. 3(d) further illustrates the temporal locations of manipulated segments, indicating that manipulations are distributed throughout the video timeline rather than being concentrated at specific positions. These statistics reflect realistic long-form scenarios where manipulations are sparse and temporally unconstrained. *(2) Diversity.* As shown in Fig. 4, TASLE is constructed from 11 distinct source datasets spanning diverse content domains, including instructional videos (*e.g.*, YouCook2 (Zhou et al., 2017)), action-centric videos (*e.g.*, FineAction (Liu et al., 2022)), and other open-world scenarios (Song et al., 2015; Miech et al., 2020). In addition, we employ three categories of controllable AI video generation paradigms (TI2V, FLF2V and MV2V) to introduce varied manipulation patterns at both segment and object levels (see Table 9 for details). This combination results in a heterogeneous dataset that captures a wide range of visual appearances, motions, and manipulation behaviors. *(3) Explainability.* TASLE provides rich linguistic annotations to support explainable forensics, containing over 28,000 rationales with a vocabulary size of 15,947 words. Fig. 3(e) and Fig. 3(f) analyze the distribution of annotated anomaly types and frequently occurring phrases in the rationales. The results show that annotations capture fine-grained anomaly categories as well as boundary-related cues (*e.g.*, *contact point/boundary anomalies* and *incongruous limb/object movement* (Ke et al., 2024; Xu et al., 2025a)), providing detailed semantic supervision for interpretable analysis of manipulated segments. These explainability annotations complement temporal localization labels and support systematic evaluation of both detection accuracy and interpretability.

**Data Splits.** The dataset is split into training and test sets, which consist of 11,179/1,293 videos, respectively. In addition, test set is further organized into multiple evaluation protocols based on criteria such as data source, AIGC generation tool, and manipulation pattern, enabling comprehensive performance analysis. Detailed descriptions of evaluation protocols are provided in Sec. 5.

# 4. Proposed MSLoc Baseline

**Problem Formulation.** Given a long untrimmed video $V = \{f_1, f_2, \ldots, f_T\}$ with $T$ frames, the goal is to identify a set of manipulated segments $\mathcal{S} = \{(t_{st}^i, t_{ed}^i, r^i)\}_{i=1}^N$, where $t_{st}^i$ and $t_{ed}^i$ denote the temporal boundaries of the $i$-th manipulated segment, and $r_i$ represents a rationale describing the visual artifacts or inconsistencies that support forensic interpretation.

## 4.1. Technical Motivation

Existing AI-generated video detectors and localization-oriented MLLMs are not well suited for the long-video, mixed real-fake setting considered in this work. We identify two fundamental challenges. ❶ *The first challenge is 'boundary-sensitive localization in mixed contexts'.* In long videos, AI-generated content is sparsely embedded within authentic footage, and the transition between real and manipulated segments often manifests as subtle boundary discrepancies. Conventional detection models are typically trained under the assumption that each input clip is entirely real or fake, making them insensitive to boundary-level cues. Meanwhile, grounding MLLMs usually rely on uniform temporal sampling when processing long videos, which dilutes the boundary information and hampers precise temporal localization. ❷ *The second challenge is 'interference from irrelevant real content'.* Untrimmed long videos contain a large amount of authentic content that is irrelevant to manipulation detection. Directly applying reasoning-heavy models using sliding-window inference leads to prohibitive computational costs, while uniform sampling introduces substantial noise from irrelevant segments. An effective solution must therefore reduce unnecessary processing while preserving sensitivity to potential manipulation regions.

Motivated by these challenges, we propose **MSLoc**, a coarse-to-fine baseline framework that decomposes long-video forensics into efficient proposal filtering followed by boundary-aware refinement. Specifically, MSLoc consists of two stages. The first stage, *Proposal Generation* (MSLoc-PG), employs a lightweight model with a sliding-window strategy to efficiently scan long videos and identify candidate segments with high manipulation likelihood, thereby filtering out the majority of irrelevant real content. The second stage, *Proposal Refinement* (MSLoc-PR), enhances a localization-oriented MLLM to focus on boundary discrepancies within the shortlisted proposals. By explicitly modeling temporal transitions and aligning multimodal manipulation cues, this stage enables reliable boundary localization together with interpretable rationale generation.

## 4.2. Baseline Design

Fig. 5 presents an overview of the proposed MSLoc baseline. Given an input long video, the framework follows a two-stage design: (1) a boundary-aware *MSLoc-PG* module for efficient proposal generation, addressing challenge ❶; and (2) a refinement-oriented *MSLoc-PR* module that performs precise boundary localization and produces interpretable rationales, addressing challenge ❷. Details are as follows.

**MSLoc-PG** focuses on efficiently identifying candidate manipulated regions from long videos while avoiding exhaustive processing of predominantly authentic content. We adopt DeMamba as the backbone for initial proposal genera-

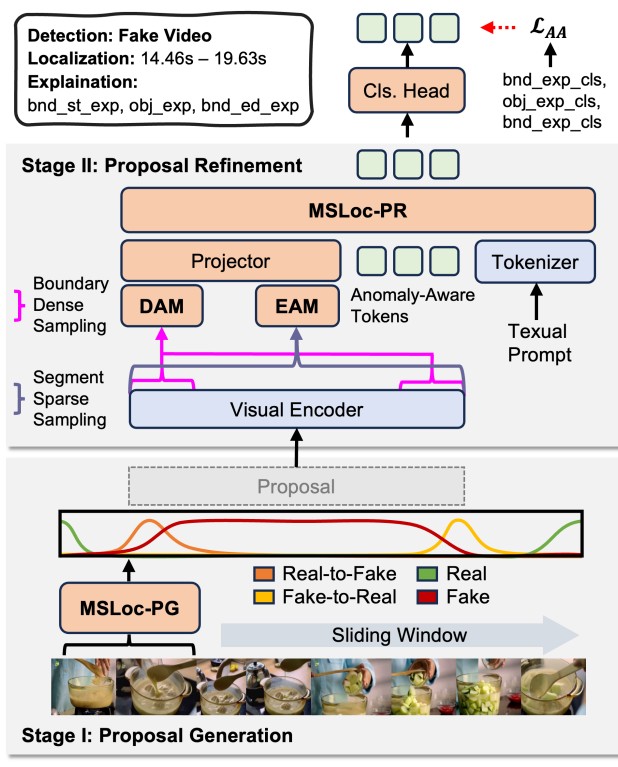

*Figure 5.* **Overall architecture of the proposed MSLoc.**

tion, as it supports end-to-end training without task-specific designs, making it a suitable and reproducible backbone for benchmarking. While DeMamba is originally designed for short video detection, we adapt it to long-video inputs using a sliding-window strategy, enabling scalable processing without sacrificing temporal coverage. Unlike DeMamba and most existing AI-generated video detectors (Zheng et al., 2025; Wen et al., 2026; Bai et al., 2025; Internò et al., 2026; He et al., 2026) that formulate detection as a binary real-versus-fake classification problem, we observe that in long-video contexts, temporal transition regions between real and manipulated content present boundary-level inconsistencies. To this end, we reformulate the detection objective as a fine-grained, boundary-aware classification problem with four classes, allowing the model to explicitly capture different temporal states around manipulation boundaries and generate more informative localization proposals. Specifically, we define a four-class label space $\mathcal{Y} = \{y_{\text{real}}, y_{\text{fake}}, y_{\text{r2f}}, y_{\text{f2r}}\}$, where $y_{\text{r2f}}$ and $y_{\text{f2r}}$ denote *real-to-fake* and *fake-to-real* boundary transitions, respectively, in addition to the standard real and fake classes $y_{\text{real}}$ and $y_{\text{fake}}$. This fine-grained formulation explicitly encourages the model to capture boundary-sensitive temporal variations and localized anomalies around manipulation transitions.

In detail, we process long videos using a sliding window of 2 seconds. Within each window, we uniformly sample 8 frames to form an input clip $X_w \in \mathbb{R}^{8 \times H \times W \times 3}$. Building

upon the proposed boundary-aware classification formulation, we train MSLoc-PG using a cross-entropy loss:

$$\mathcal{L}_{\text{ce}} = -\frac{1}{N_b} \sum_{i=1}^{N_b} \log(p_{i,t_i}), \quad (1)$$

where $N_b$ is the batch size, $t_i$ is the ground-truth class index for sample $i$, and $p_{i,t_i}$ is the predicted probability of the true class. Once trained, the model processes the video in a non-overlapping sliding window fashion. We merge consecutive windows with positive detections (including boundary classes) to form the initial proposal set $\mathcal{P} = \{P_1, P_2, \ldots, P_M\}$, where each proposal $P_i$ represents a coarse temporal region. These proposals are subsequently passed to the refinement stage for precise localization and explanation.

**MSLoc-PR** aims to refine coarse proposals and generate evidential explanations using an MLLM. Given a coarse proposal $P_i$, we apply adaptive sampling over two semantically distinct region types: a *boundary region* and an *event region*. The boundary region (the leftmost and rightmost $\phi\%$ regions of the proposal) focuses on temporal transitions between real and manipulated content. Since precise localization relies on capturing subtle boundary cues, we perform *dense sampling* in this region ($2 \times N_b$ frames) to model fine-grained temporal variations. In contrast, the event region corresponds to the interior of the manipulated segment, where the presence of anomalies has already been established. For this region, we apply *sparse sampling* (8 frames) to extract high-level semantic context for explanation generation. The sampled frames are subsequently encoded by a visual encoder, producing region-specific visual features $F_b$ and $F_e$.

To enable efficient and boundary-sensitive refinement, we introduce two dedicated feature processing modules: *Difference-Aware Modeling* (**DAM**) and the *Event Aggregation Module* (**EAM**). DAM operates on the boundary features $F_b$ by employing a Q-Former (Guo et al., 2024) to compress frame-level tokens, thereby condensing dense temporal information. Meanwhile, leveraging the similarity prior between spatially corresponding tokens in adjacent frames, we derive inter-frame variant and inter-frame invariant tokens through weighted averaging, and feed them into the MLLM decoder. This design effectively preserves more discriminative transition cues. EAM is applied to the event features to aggregate semantic information within the manipulated segment. We employ a spatio-temporal joint token compression strategy (Guo et al., 2024) to reduce the token sequence length, thereby minimizing the computational overhead of the MLLM decoder while retaining high-level semantic context for explanation generation.

To further enhance the model's comprehension of diverse manipulation cues and its capability in explanation reason-

ing, we introduce three anomaly-aware tokens as input. These tokens are encoded by LLM to capture anomalous cues within the video. Subsequently, the output embeddings of these tokens are fed into a classification head to predict anomaly categories. These categories (*e.g.*, boundary explanation class, denoted as `bnd_exp_cls`) correspond to the rationales in the explanation ground truth (*e.g.*, boundary start explanation, denoted as `bnd_st_exp`). The tokens are optimized using the cross-entropy loss, which we term the Anomaly-Aware Loss (denoted as $\mathcal{L}_{AA}$).

# 5. Experiments

## 5.1. Evaluation Metrics

**Detection and Localization:** In long-video forensic scenarios, conventional metrics such as mean Average Precision (mAP) are inadequate for two main reasons: (1) mAP relies on confidence-ranked predictions to compute precision-recall curves, whereas MLLM-based models directly output temporal coordinates without reliable confidence scores; (2) mAP does not explicitly penalize false positive predictions on fully authentic videos, which is critical in forensic settings where authentic videos are prevalent. To address these issues, we introduce $F1_{Det}$, which divides each video into fine-grained 0.01s detection windows and computes the F1 score over all windows, thereby directly penalizing false positives on authentic content. For temporal localization, following E.T. Bench (Liu et al., 2024), we adopt $F1_{Loc}$, which calculates the average retrieval F1 score under the IoU threshold set $\{0.1, 0.3, 0.5, 0.7\}$ based on the retrieval matching of localized segments to reflect localization completeness.
**Rationale Quality:** We employ GPT 5.1 as scorer to evaluate the matching degree between the model-generated rationale for AIGC segments and the ground truth annotations.
**Generalization Evaluation:** We further evaluate the model's generalization to unseen generation techniques. A subset of AIGC tools (*e.g.*, Kling (Team et al., 2025) and Jimeng (ByteDance, 2024)) is involved from training and treated as *unseen AIGC types* for testing. In addition, to assess robustness under distribution shift, we conduct realistic manual malicious manipulations on videos from the TVSum dataset (Song et al., 2015), which is entirely excluded from training, and regard this setting as *out-of-domain data*.

## 5.2. Implementation Details

The proposed MSLoc framework is trained in a two-stage manner. In the first stage, MSLoc-PG is trained using a sliding-window strategy with a window duration of 2 seconds (8 frames), optimized under the proposed boundary-sensitive classification objective. For existing detection methods such as BusterX++ (Wen et al., 2026), we follow their default training protocols and optimize them using the standard binary classification objective. For the second stage, MSLoc-PR employs a Trace (Guo et al., 2024)-based MLLM for localization and rationale generation. Following the fine-tuning protocol of TRACE, we train MSLoc-PR for 2 epochs with a batch size of 16 and a learning rate of $3 \times 10^{-5}$. For localization models, we support both two-stage and end-to-end training by taking either proposals generated by MSLoc-PG or complete long videos as input. We sample 10% of the training set as the validation set. All experiments are conducted on a cluster of 8 H100 GPUs.

## 5.3. Main Results

**Detection Models with Sliding Window Strategy.** We select three representative detection models for comparison: the training-free short video detection model D3, the MLLM BusterX++ designed for AI-generated short video detection, and the general MLLM Qwen3-VL-8B. As shown in Table 2, these models perform poorly when directly transferred to long video scenarios. After fine-tuning on our TASLE dataset (denoted as *), the performance of all models improves, indicating that the dataset provides effective supervision signals for manipulated segments localization. Furthermore, comparing with the binary-classification-based DeMamba, MSLoc-PG outperforms across all metrics using the same backbone architecture. This demonstrates that modeling the task as a fine-grained boundary classification (including Real-to-Fake and Fake-to-Real) helps the model capture anomalies at the edges of generated segments, thereby improving localization precision.

**Localization Models.** We compare MSLoc with the mainstream localization model Trace trained on our dataset. Constrained by uniform frame sampling, end-to-end localization LLMs are susceptible to interference from a vast amount of irrelevant video segments. This impairs both their localization capability and the explanation generation capability for AI-generated segments, rendering them inferior to small-scale models based on sliding-window inference. However, leveraging the proposals predicted by DeMamba and MSLoc-PG allows the large model Trace to effectively circumvent the interference of irrelevant segments. By concentrating on fine-tuning for anomaly localization and explanation generation, it achieves significant improvements. Furthermore, our MSLoc-PR incorporates a specific focus on boundary regions, yielding superior results and demonstrating the superiority of the two-stage paradigm. Notably, the improvement brought by the MLLM-based refinement stage is more pronounced on out-of-domain data (+7.8 $F1_{Det}$, +17.6 $F1_{Loc}$ over MSLoc-PG alone), suggesting that the MLLM captures more transferable semantic cues compared to the Mamba-based proposal generator.

**Explainability Analysis.** We evaluate the Rationale Quality (RQ) of the explanations generated by localization LLMs

*Table 2.* **Quantitative evaluation on the test set of TASLE dataset**. * represents training with our dataset.

| Method | In-domain Data | | | | | | Out-of-Domain Data | | |
|---|---|---|---|---|---|---|---|---|---|
| | Seen AIGC Type | | | Unseen AIGC Type | | | Mixed AIGC Type | | |
| | $\mathbf{F1}_{Det}$ | $\mathbf{F1}_{Loc}$ | RQ | $\mathbf{F1}_{Det}$ | $\mathbf{F1}_{Loc}$ | RQ | $\mathbf{F1}_{Det}$ | $\mathbf{F1}_{Loc}$ | RQ |
| *Detection Models (w/ sliding window strategy)* | | | | | | | | | |
| D3 (Zheng et al., 2025) | 34.6 | 31.1 | ✗ | 38.5 | 30.4 | ✗ | 27.4 | 22.5 | ✗ |
| BusterX++ (Wen et al., 2026) | 8.3 | 7.8 | ✗ | 3.6 | 1.5 | ✗ | 4.2 | 4.6 | ✗ |
| BusterX++* | 33.6 | 36.4 | ✗ | 39.0 | 34.1 | ✗ | 27.8 | 26.9 | ✗ |
| Qwen3-VL-8B (Bai et al., 2025) | 8.1 | 8.1 | ✗ | 8.4 | 7.2 | ✗ | 7.1 | 6.2 | ✗ |
| Qwen3-VL-8B* | 27.6 | 21.8 | ✗ | 29.9 | 19.9 | ✗ | 22.9 | 18.0 | ✗ |
| DeMamba* (Chen et al., 2024) | 54.9 | 54.0 | ✗ | 52.5 | 38.7 | ✗ | 40.8 | 33.0 | ✗ |
| **Our MSLoc-PG** | **67.5** | **64.8** | ✗ | **62.7** | **50.0** | ✗ | **47.2** | **38.7** | ✗ |
| *Localization Models* | | | | | | | | | |
| Trace* (Guo et al., 2024) | 42.0 | 41.4 | 3.32 | 38.7 | 38.0 | 3.47 | 31.8 | 32.4 | 3.11 |
| Trace* (w. DeMamba*) | 55.7 | 59.1 | 3.45 | 56.3 | 56.5 | 3.67 | 50.8 | 52.0 | 3.27 |
| Trace* (w. MSLoc-PG) | 69.0 | 70.9 | 3.91 | 63.5 | 62.1 | 3.74 | 52.7 | 54.0 | 3.67 |
| **Our MSLoc** | **70.1** | **72.2** | **4.05** | **67.0** | **62.8** | **4.04** | **55.0** | **56.3** | **3.88** |

*Table 3.* **Ablation analysis of our MSLoc-PR**. Here backbone network stands for the Trace with MSLoc-PG.

| DAM | EAM | $\mathcal{L}_{\mathbf{AA}}$ | $\mathbf{F1}_{Det}$ | $\mathbf{F1}_{Loc}$ | RQ |
|---|---|---|---|---|---|
| Backbone Network | | | 61.7 | 62.3 | 3.77 |
| ✓ | | | 62.8 | 63.0 | 3.56 |
| ✓ | ✓ | | 63.5 | 63.2 | 3.79 |
| ✓ | ✓ | ✓ | **64.0** | **63.8** | **3.99** |

*Table 5.* **Efficiency analysis of different models**.

| Method | Time (m) | $\mathbf{F1}_{Det}$ | $\mathbf{F1}_{Loc}$ | RQ |
|---|---|---|---|---|
| BusterX++ | 25 | 33.5 | 32.5 | ✗ |
| DeMamba | 2 | 49.4 | 41.9 | ✗ |
| MSLoc-PG | 2 | 59.1 | 51.2 | ✗ |
| Trace | 9 | 37.5 | 37.3 | 3.30 |
| MSLoc | 12 | **64.0** | **63.8** | **3.99** |

*Table 4.* **Ablation on the impact of hyperparameters of DAM**.

| $\phi\%$ | $N_b$ | $\mathbf{F1}_{Det}$ | $\mathbf{F1}_{Loc}$ | RQ |
|---|---|---|---|---|
| 20 | 8 | 63.2 | 63.0 | 3.76 |
| 10 | 16 | 63.8 | **64.0** | 3.80 |
| 30 | 16 | 63.6 | 63.7 | **4.01** |
| 20 | 16 | **64.0** | 63.8 | 3.99 |

on a scale of 0 to 5. As shown in Table 2, compared to the end-to-end localization paradigm of Trace, two-stage methods yield higher RQ scores. Furthermore, the RQ improves as the localization quality of the proposals predicted by the first-stage model increases. This indicates that the key to generating high-quality explanations lies in the model's focus on AI-generated segments. Our MSLoc achieves the best RQ, demonstrating that the second-stage localization model's attention to boundary information and its understanding of anomaly cues are also critical.

**Generalization Analysis.** We conducted tests on Out-of-Domain Data and Unseen AIGC Types scenarios. The results indicate that insufficient generalization capability is a common issue among models, and out-of-domain data may pose even greater challenges.

### 5.4. Ablation Study

**Contribution of Key Components.** We evaluate the contributions of the DAM, EAM, and $\mathcal{L}_{AA}$ within MSLoc-PR. Compared to the baseline (uniform frame sampling for proposals), the introduction of DAM enables the model to successfully focus on boundary regions, thereby improving localization results. However, the loss of event content compromises the quality of rationales. The incorporation of $\mathcal{L}_{AA}$ further stimulates the model's perception of anomaly cues and enhances the generation quality.

**Boundary Range and Sampling Frame Count.** We investigate the impact of the boundary region (the leftmost and rightmost $\phi\%$ regions of the proposal) and the number of frames sampled within this region ($2 \times N_b$). The experimental results in Table 4 indicate that reducing frame sampling leads to a decline in both localization ($\mathbf{F1}_{Det}$, $\mathbf{F1}_{Loc}$) and interpretability (RQ), underscoring the importance of dense boundary frame sampling. For the boundary range, the model exhibits stable localization performance within $\phi \in [10\%, 30\%]$, with the best localization trade-off achieved at $\phi = 20\%$. Notably, RQ improves as $\phi$ increases from 10% to 30% (3.80 → 4.01), suggesting that a moderately broader boundary context provides richer semantic information for explanation generation. However, this bene-

fit does not extend to localization, as overly broad regions risk introducing redundant authentic content that dilutes discriminative boundary cues.

**Efficiency Analysis.** We compare the computational efficiency of our MSLoc and MSLoc-PG with other models in processing long videos. All experiments are conducted on a cluster of 8 NVIDIA H100 GPUs. We report the total inference time (in minutes) on the TASLE test set. As shown in Table 5, the detection LLM BusterX++ suffers from slow inference speeds. While Mamba-based detection models (i.e., DeMamba, MSLoc-PG) are efficient, they lack the capability to generate explanations. Compared to localization LLMs Trace, our MSLoc achieves remarkable localization performance and rationale quality with low additional computational overhead, demonstrating the accuracy advantage of our two-stage approach.

## 6. Discussion and Outlook

While MSLoc achieves promising results in long-video forgery localization, certain limitations and open directions remain. First, due to its cascaded coarse-to-fine architecture, the final performance is bounded by the recall of the proposal generation stage; segments missed initially cannot be recovered by the subsequent MLLM. Future work could explore end-to-end joint training mechanisms to mitigate such error propagation. Second, the model's detection capability relies on the visibility of generative artifacts. As video generation technologies evolve rapidly, visual artifacts are becoming increasingly subtle. We plan to continuously update TASLE with outputs from emerging generators and to extend our research to more challenging manipulation types, such as leveraging physical constraints (Zhang et al., 2026) or global semantic consistency, to tackle the highly realistic but logically incoherent forgeries emerging from advanced generation tools. Additionally, incorporating detection signals beyond visual appearances, such as audio-visual synchronization and physics-aware reasoning (e.g., gravity consistency, motion dynamics), represents a promising direction for future work.

## 7. Conclusion

In this work, we study the problem of detecting and explaining AI-generated manipulations in untrimmed long videos. We introduced a new long-video forensic setting, together with a large-scale benchmark, TASLE, and a baseline framework, MSLoc, to support systematic evaluation under challenging mixed real–fake scenarios. Experimental results highlight the importance of boundary localization and interpretable reasoning for long-form video forensics. We hope this work will facilitate future research toward more robust and trustworthy analysis of AI-generated videos.

## Acknowledgement

This work was partially supported by New Generation Artificial Intelligence-National Science and Technology Major Project (No. 2025ZD0122903), the National Natural Science Foundation of China (No. U25A20533, No. 62276129, No. 62506165), the Natural Science Foundation of Jiangsu Province (No. BK20250082), the Fundamental Research Funds for the Central Universities (No. NE2025010, No. NS2025038), the Jiangsu Funding Program for Excellent Postdoctoral Talent (No. 2025ZB306), the Outstanding Doctoral Dissertation in NUAA (No. BCXJ25-24).

## Impact Statement

Our work has a positive impact on society by advancing the field of trustworthy AI through the development of explainable forensic techniques for detecting manipulated segments in long-form videos. By introducing a dedicated benchmark and a baseline method, we promote research into robust and interpretable video authenticity analysis, which is crucial for combating misinformation in an era of increasingly realistic AI-generated content. We will open-source the proposed dataset and source code to facilitate reproducibility and encourage future research on this task.

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

# A. TASLE Dataset

This section provides detailed information on the composition and characteristics of the TASLE dataset. As mentioned in the main text, TASLE is specifically designed for the task of localizing and explaining sparse AI-generated segments in long videos, encompassing a diverse range of video content and manipulation patterns. Visual examples are illustrated in Fig. 6, which showcases the richness and variety of content in our dataset, spanning multiple scenarios and manipulation types.

## A.1. Data Sources and Diversity

To ensure the diversity and representativeness of the dataset, we collected long-form videos from 11 publicly available sources, covering a wide variety of scenarios and content types, as detailed in Table 8. These video sources include cooking tutorials (Youcook2 (Zhou et al., 2017)), fine-grained human actions (FineAction (Liu et al., 2022)), desktop activities (GTEA (Fathi et al., 2011)), kitchen operations (EK100 (Damen et al., 2022)), industrial scenarios (ENIGMA-51 (Ragusa et al., 2024)), animal behavior (MammAlps (Gabeff et al., 2025)), rare actions (RareAct (Miech et al., 2020)), anomaly behavior monitoring (UCA (Yuan et al., 2023)), diverse categories (TVSum (Song et al., 2015)), traffic surveillance (I24V (Gloudemans et al., 2024)), and pedestrian/vehicle scenes (TAO (Dave et al., 2020)). The video durations range from 2 seconds to 124 seconds, encompassing various real-world scenarios such as first-person, third-person perspectives, and indoor/outdoor settings. Table 6 and 7 provide the statistical results of the propose TASLE dataset.

## A.2. Manipulation Patterns and Generation Methods

The TASLE dataset incorporates three types of controllable AI video generation paradigms: FLF2V (First-Last-Frame-to-Video), TI2V (Text-Image-to-Video), and MV2V (Mask-Video-to-Video), simulating video manipulations at different granularities and approaches. As shown in Table 9, each AIGC tool exhibits certain limitations in terms of generated duration. To construct data that aligns with the sparse-manipulation characteristics of long videos, we adopt flexible segment-processing strategies: for scenarios requiring longer segments, multiple generated clips are temporally concatenated; for scenarios requiring shorter segments, down-sampling or cropping operations are applied.

# B. Dataset Construction Pipelines

This section elaborates on the two core human-in-the-loop pipelines in the TASLE dataset construction process: the Data Processing Pipeline for generating manipulated videos and the Rationale Annotation Pipeline for producing ex-

*Table 6.* **Statistics of the proposed TASLE.**

| * | Train | Test | Total |
|---|---|---|---|
| Videos | 11,179 | 1,293 | 12,472 |
| AIGC Segments | 5,970 | 695 | 6,665 |
| Video Duration (Hours) | 80.2 | 8.1 | 88.3 |
| AIGC Duration (Hours) | 7.2 | 0.8 | 8.1 |
| Segment-level Rationales | 16,229 | 1,537 | 17,766 |
| Boundary-level Rationales | 9,595 | 794 | 10,389 |

*Table 7.* **Statistics of rationales within TASLE.**

| * | Count |
|---|---|
| Rationales | 28,155 |
| Vocabulary Size | 15,947 |
| Avg. Words per Rationale | 28.6 |
| Avg. Rationales per AIGC Segment | 4.2 |

planatory annotations.

## B.1. Data Processing Pipeline

To realistically simulate the scenario of locally embedded AI-generated content within long videos, we designed a controlled protocol for video segment creation and insertion. The process is illustrated in Fig. 7.

For **segment-level generation**, candidate video segments corresponding to salient actions or events are identified based on temporal localization annotations from the source datasets. To avoid overfitting to the original segment distribution of the source datasets, we perform random segment selection and temporal clipping. Subsequently, a large language model is employed to generate misleading event semantic descriptions for the selected segments. Based on the resulting textual descriptions and reference frames extracted from the candidate segments (the first and last frames for the FLF2V paradigm, and a single frame for the TI2V paradigm), corresponding AIGC tools are used to generate replacement AI video segments. For generators lacking explicit control over ending frames, shot detection is applied to ensure temporal coherence between the generated segment and the subsequent real video.

For **object-level generation**, spatio-temporal annotations from the source datasets (e.g., bounding boxes from TAO and I24V datasets) are leveraged to localize salient objects (e.g., pedestrians, vehicles) within the videos. We apply SAM2-based video object segmentation (Ravi et al., 2024) to obtain object masks (Li et al., 2025a; Ji et al., 2023; 2024a;b; Zhao et al., 2024; Wu et al., 2025), filtering out objects that are too small or persist for excessively long durations. The selected targets are then replaced or removed using a mask-conditioned video-to-video generation tool (e.g., VACE), producing fine-grained object-level manipulations.

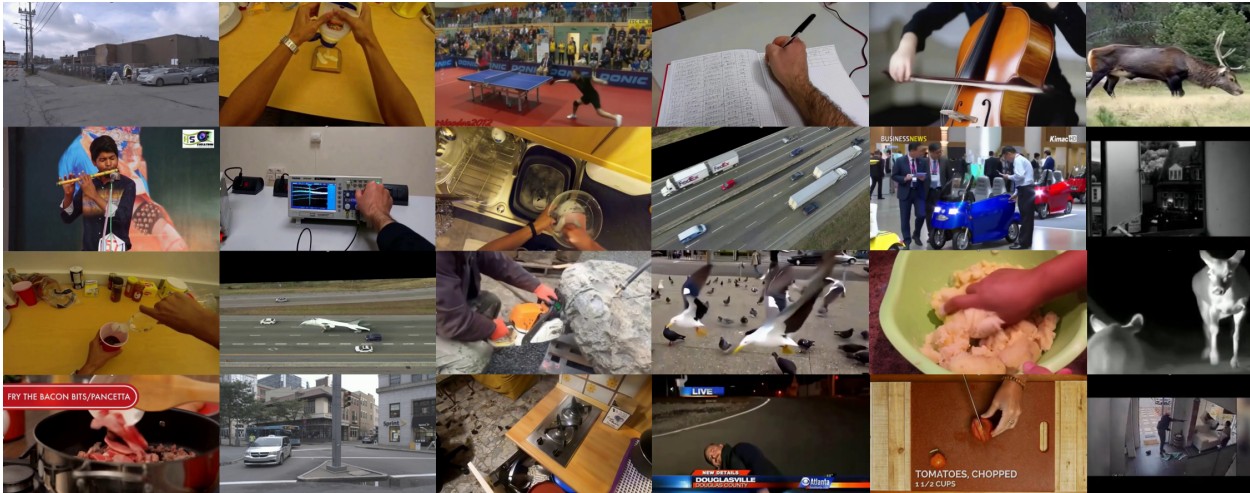

*Figure 6.* **Examples from the TASLE dataset**. For more samples, please visit our project website via this link.

The generated AI content is seamlessly integrated back into the original long videos. All videos are standardized to a resolution of 832×480 at 15 FPS. Automatic consistency checks are then performed, including verification of resolution, frame count, and temporal alignment. Finally, human inspection by six annotators is conducted to verify visual quality and the smoothness of transitions between real and AI-generated segments, resulting in realistic and challenging manipulation cases.

### B.2. Rationale Annotation Pipeline

To support explainable forensics, TASLE provides boundary-level and object-level rationales for each manipulated segment. The annotation pipeline, shown in Fig. 8, faces the core challenge of the high realism of the AI-generated content in our dataset. Therefore, we adopt a comparative annotation strategy to ensure accuracy and targeted focus.

Concretely, for each AI-generated segment, the annotation model (Qwen3-VL-235B) is provided with two sets of materials: (1) A video pair for comparison. This includes the AI-generated segment and the corresponding reference segment extracted from the original real video (or the pre-masked video for object-level generation). To facilitate fine-grained spatio-temporal alignment, the reference video and the AI-generated video are vertically concatenated, enabling direct frame-wise comparison. (2) Boundary context. For boundary-level rationales, an additional video segment containing the transition regions before and after the manipulation start and end points is provided.

Then, using carefully designed prompts, we guide the large model to focus, through comparative analysis, on discriminative visual differences between the real and AI-generated content, rather than speculative artifacts. The prompts ex-plicitly instruct the model to describe anomalies at the transitions (boundary-level) or visual flaws within the generated content (object/segment-level). Based on the provided video materials and prompts, the annotation model generates natural language rationales describing the inconsistencies. Finally, all model-generated rationales undergo manual screening and correction by six inspectors to ensure salience, accuracy, and consistency, ultimately forming high-quality annotation results.

**Human Verification Statistics.** During manual screening, six trained annotators evaluated each rationale using five hierarchical criteria: (1) visual object correctness, (2) manipulation type accuracy, (3) object-description consistency, (4) description clarity, and (5) format completeness. Errors were counted hierarchically, i.e., if an earlier category failed, later categories were not additionally counted. Across 28,155 rationale samples, the most frequent issue was incorrect visual object identification (12.3%), including misidentified or omitted key objects (e.g., an original rationale describing a "facial" anomaly was corrected to "hair motion" upon inspection). Manipulation type errors accounted for 9.8% (e.g., describing a motion anomaly as a texture issue), followed by insufficient description clarity (5.4%), object-description inconsistency (2.7%), and format or truncation issues (1.2%). These results indicate that model-generated rationales primarily struggle with accurate object identification, motivating the human verification step in our annotation pipeline.

## C. Experiments

### C.1. Visualization Results

Fig. 9 presents visualization results of our method on the task of local manipulation detection and explanation in long

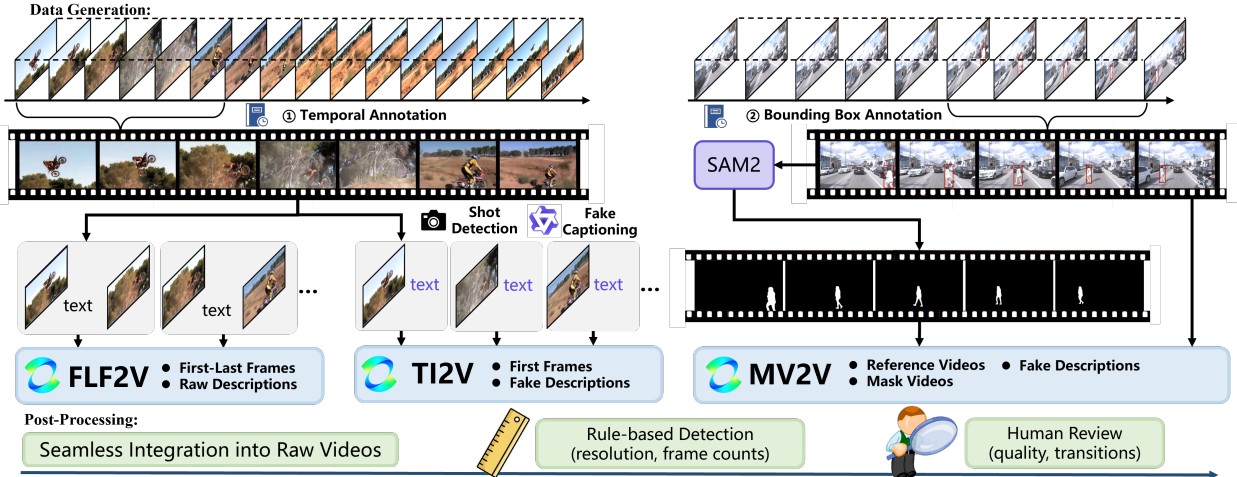

*Figure 7.* **Overview of our human-in-the-loop dataset processing pipeline.**

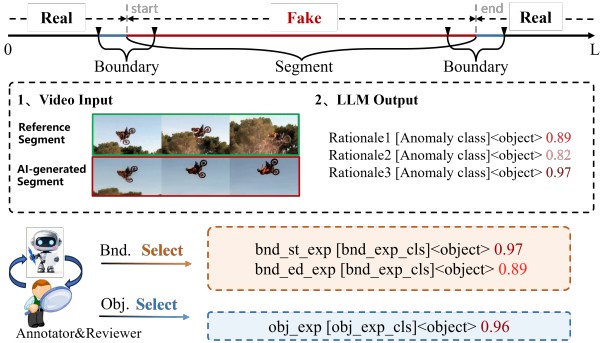

*Figure 8.* **Rationale annotation pipeline for TASLE.**

videos. As shown, our model not only accurately localizes the manipulated temporal intervals (highlighted in red), but also generates detailed explanatory texts for each interval, clearly identifying the specific objects involved (e.g., "hand motion") and the types of anomalies (such as "rigid movement" or "inconsistent occlusion"). Moreover, our model is able to capture boundary changes and abrupt shifts in motion inertia at the points of manipulation, demonstrating a deep understanding of temporal context. The generated explanations are highly consistent with the actual manipulations, providing strong evidence for tampering detection and provenance analysis, and fully validating the effectiveness and practical value of our approach in explainable video forensics.

## D. Discussion: Difference from Deepfake Detection

While both Deepfake detection and the proposed Temporal AI-Generated Segment Localization and Explaination task aim to identify synthesized content, they differ fundamen-

tally in scope, modality, and technical challenges.

**Target Scope and Priors.** Deepfake detection primarily focuses on facial manipulation, including identity swapping (DeepFakes), expression reenactment (Face2Face), and lip-syncing. These methods often rely on strong face-specific priors, such as facial landmarks and biological signals (e.g., blinking patterns). In contrast, our task targets general AI-generated videos (e.g., produced by text-to-video models like Sora or Wan). The manipulation extends beyond faces to include open-world scenes, objects, backgrounds, and physical dynamics. Consequently, face-centric priors are ineffective, necessitating models that can reason about general scene semantics, lighting consistency, and physical plausibility.

**Modality and Cues.** Deepfake detection frequently utilizes multi-modal cues, checking for synchronization between audio and visual streams (e.g., lip movements matching speech). Our current benchmark focuses on visual forensics in long untrimmed videos. The core challenge lies in detecting purely visual artifacts, such as unnatural temporal transitions, spatial distortions in complex backgrounds, and inconsistencies in object interactions.

**Generation Paradigms.** Deepfake content is typically generated using GANs or Autoencoders tailored for facial textures. The resulting artifacts often manifest as blending boundaries on the face or resolution mismatches. The AIGC segments in TASLE are produced by state-of-the-art generating models (e.g., FLF2V, TI2V). These models generate high-fidelity textures but may exhibit hallucinations, temporal flickering, or logical inconsistencies over time, requiring different detection strategies as explored in our MSLoc framework.

*Table 8.* **Overview of the sourced video datasets used in TASLE.**

| Dataset | Video Content | Annotation | Type of Tools | AIGC Tools | Open-source Policies |
|---|---|---|---|---|---|
| Youcook2 (Zhou et al., 2017) | Cooking Tutorial | Temporal | TI2V FLF2V | LTXVideo, SkyReels-V1, hailuo, Wan2.1, vidu, kling, jimeng | MIT License |
| FineAction (Liu et al., 2022) | Human Actions, Third-Person | Temporal | TI2V FLF2V | LTXVideo, SkyReels-V1, hailuo, Wan2.1, vidu, kling, jimeng | MIT License |
| GTEA (Fathi et al., 2011) | Desktop, First-Person | Temporal | FLF2V | Wan2.1, vidu, jimeng | MIT License |
| EK100 (Damen et al., 2022) | Kitchen, First-Person | Temporal | FLF2V | Wan2.1, vidu, kling, jimeng | CC BY-NC-SA 4.0 |
| ENIGMA-51 (Ragusa et al., 2024) | Industrial Operations, First-Person | Temporal | FLF2V | Wan2.1, vidu, jimeng | CC BY 4.0 |
| MammAlps (Gabeff et al., 2025) | Animals, Third-Person | Temporal | TI2V | hailuo, kling | MIT License |
| RareAct (Miech et al., 2020) | Rare Actions | Temporal | TI2V FLF2V | LTXVideo, SkyReels-V1, hailuo, Wan2.1, vidu, kling, jimeng | Apache License 2.0 |
| UCA (Yuan et al., 2023) | Anomaly Behavior Monitoring | Temporal | FLF2V | vidu, kling | Apache License 2.0 |
| TVSum (Song et al., 2015) | Diverse Categories | Temporal | TI2V FLF2V | LTXVideo, SkyReels-V1, hailuo, Wan2.1, vidu, kling | MIT License |
| I24V (Gloudemans et al., 2024) | Traffic Surveillance, Third-Person | Bounding box | MV2V | VACE | MIT License |
| TAO (Dave et al., 2020) | Vehicle/Pedestrian, First-Person | Bounding box | MV2V | VACE | MIT License |

*Table 9.* **Details of the AIGC tools used in TASLE.**

| Type | AIGC Tools | Capability of Model Generation | Duration of AIGC Segments | Out-of-Domain | Open Source |
|---|---|---|---|---|---|
| FLF2V | Wan2.1 (Wan et al., 2025) | Up to 5s, variable length | 3-13.4s | ✗ | ✓ |
| | kling (Team et al., 2025) | Fixed length of 5s | 5.4-15s | ✓ | ✗ |
| | vidu (Bao et al., 2024) | Fixed length of 4s | 4-5s | ✗ | ✗ |
| | jimeng (ByteDance, 2024) | Fixed length of 5s | 5.7-14.5s | ✓ | ✗ |
| TI2V | LTXVideo (HaCohen et al., 2024) | Up to 5s, variable length | 3-7s | ✗ | ✓ |
| | SkyReels-V1 (SkyReels-AI, 2025) | Up to 5s, variable length | 3-9.2s | ✗ | ✓ |
| | hailuo (MiniMax, 2024) | Fixed length of 6s | 3-5.2s | ✗ | ✗ |
| | kling (Team et al., 2025) | Fixed length of 5s | 3-5.8s | ✓ | ✗ |
| MV2V | VACE (Jiang et al., 2025b) | Fixed length of 5s | 0.2-7.6s | ✗ | ✓ |

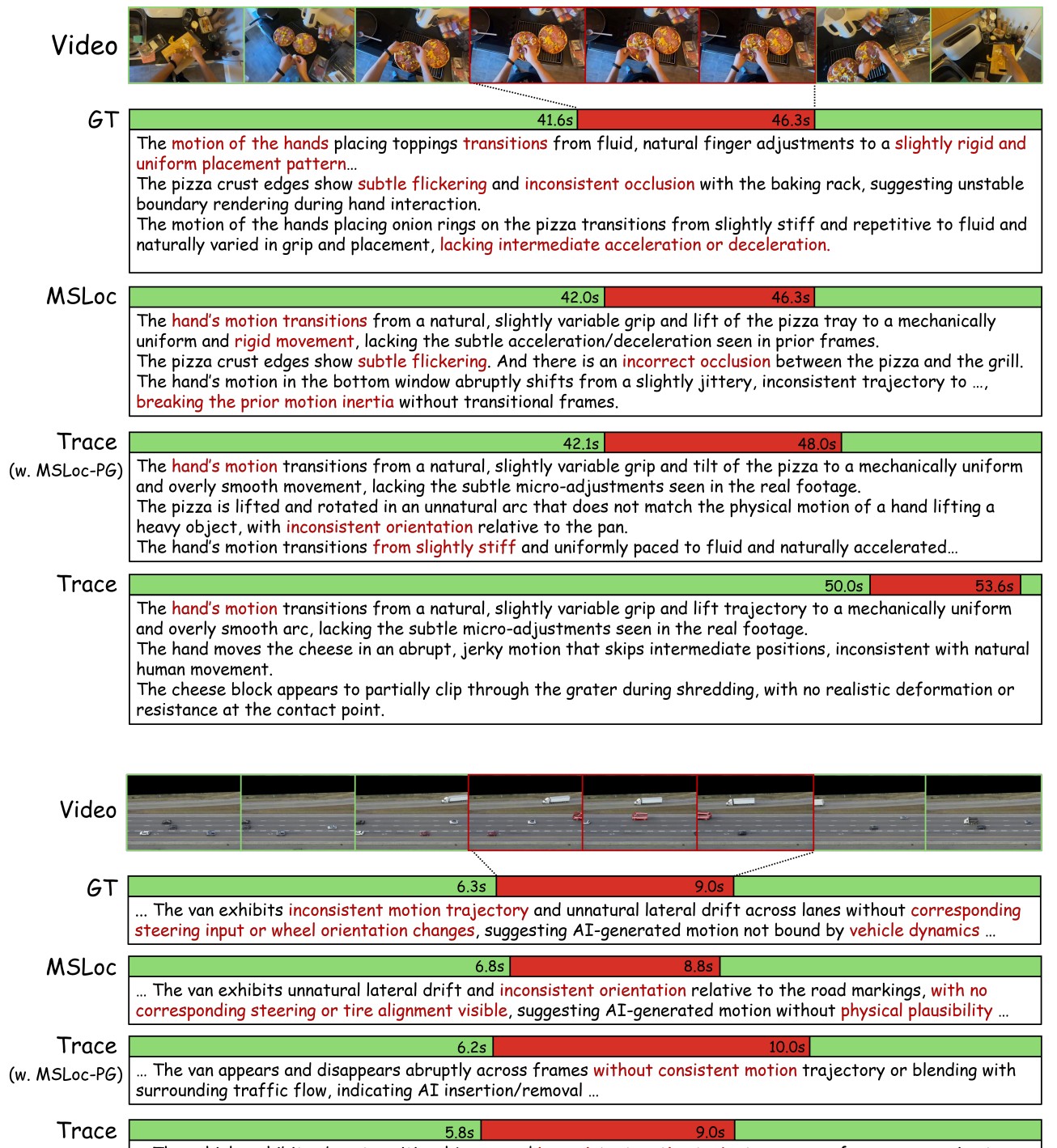

*Figure 9.* **Visual results of long-video AI-generated segment localization and explanation.** Red segments indicate the temporal intervals detected as manipulated by the model, while green segments denote authentic content. Below each interval, the model-generated explanatory texts are shown, detailing the specific objects and types of anomalies (e.g., "hand motion", "inconsistent occlusion").

