# OpenReview forum: "Explainable Forensics of Manipulated Segments in Untrimmed Long Videos"
_ICML.cc/2026/Conference — ICML 2026 regular_

### Official Review · Reviewer_RLFM · 2026-02-26

**Soundness:** 2
**Presentation:** 3
**Significance:** 3
**Originality:** 4
**Overall Recommendation:** 4
**Confidence:** 5

**Summary:**

This paper addresses the problem of Temporal AI-Generated Segment Localization and Explanation (TASLE) in untrimmed long videos. The authors introduce a large-scale benchmark, TASLE, containing 12,472 videos with segment-level and object-level manipulations, accompanied by rich forensic rationales. They propose MSLoc, a coarse-to-fine baseline that utilizes a sliding-window Proposal Generation module (MSLoc-PG) and an MLLM-based Proposal Refinement module (MSLoc-PR). Experimental results demonstrate that MSLoc outperforms existing short-video detection methods when adapted to the long-video setting.

**Compliance With Llm Reviewing Policy:**

Affirmed.

**Final Justification:**

The authors have addressed my concerns. I believe this benchmark represents a solid contribution and a promising step forward for the field of AIGV forensics.

**Key Questions For Authors:**

See weaknesses.
The motivation and starting point of this paper are commendable. However, the massive scope of the task including data production, complex labeling, and new framework design has led to the omission of several critical technical details (as noted in the Major Weaknesses). If the authors can provide more detailed explanations or additional data addressing these Major Weaknesses, I would be happy to raise my score to a 4 or higher.

**Limitations:**

See weaknesses.

**Strengths And Weaknesses:**

Strengths:
1. Shifts the focus from binary short-clip classification to the more realistic scenario of sparse manipulations in long-form videos.
2. The TASLE dataset provides diverse manipulation patterns (FLF2V, TI2V, MV2V) and high-quality linguistic annotations for explainability.

Minor Weaknesses:
1. In fig. 2 TI2V workflow, the diagram appears to show two `first frames'. Is the second `first frame' also a real frame? And whether the baseline can simultaneously detect and localize multiple non-contiguous manipulated segments within a single long video?
2. Object-level manipulations (MV2V) generally appear more photorealistic than segment-level replacements. The authors should provide more comparative data on the performance gap between detecting these two distinct types of forgeries.
3. For Fig.5, I suggest swapping the vertical positions of Stage I and Stage II .

Major Weaknesses:
1. The `comparative annotation strategy' described on page 4 is presented as a core contribution I think to solving the challenges of video explanation. However, this strategy is already widely used in image-level explainability work [1][2]. The authors fail to provide a sufficiently detailed process of how this strategy work and how to adapt the temporal complexities of video forensics.
2. The paper lacks a deep dive into failure cases. There is no analysis of the average localization error or whether inconsistencies exist between localization accuracy and rationale quality. Furthermore, an error analysis of the generated linguistic output (regarding format and content accuracy) is missing.
3. It is unclear whether the baseline model develops a general understanding of videos or if it simply learns a strong correlation mapping specific to the training data distribution.
4. The authors adopt DeMamba as the backbone for proposal generation. DeMamba is no longer considered SOTA in video detection [3][4][5]. The authors should provide DeMamba’s initial performance on TASLE and discuss whether its limitations lead to the truncation or complete omission of manipulated segments during the sliding-window phase.
5. A key factor in localization performance is the specific percentage of the `leftmost and rightmost' regions selected for the boundary. More experiments are required to justify why specific percentages (e.g., 10%-20%) are optimal.

[1] Unlocking the Capabilities of Large Vision-Language Models for Generalizable and Explainable Deepfake Detection.

[2] VLForgery Face Triad: Detection, Localization and Attribution via Multimodal Large Language Models

[3]  Ai-generated video detection via perceptual straightening

[4] Physics-driven spatiotemporal modeling for ai-generated video detection

[5] MPF-Net: Exposing High-Fidelity AI-Generated Video Forgeries via Hierarchical Manifold Deviation and Micro-Temporal Fluctuations

---

> ### Author Rebuttal · Authors · 2026-03-31
>
> Dear Reviewer RLFM, thank you for your comments. We respond below.
>
> ---
>
> **Q1 (Minor 1):** *...two first frames...whether baseline can detect multiple non-contiguous segments...*
>
> (1) Yes, the second "first frame" is also real, used as the conditioning reference for the next TI2V step. We apply TI2V repeatedly at shot boundaries to extend generation while preserving transitions.
>
> (2) Yes. MSLoc-PG scans via sliding windows and merges consecutive positives into proposals {P1,...,PM}, naturally yielding multiple disjoint segments. In our benchmark, 11.7% of videos contain ≥2 non-contiguous manipulated segments.
>
> ---
>
> **Q2 (Minor 2):** *...comparative data on performance gap between object-level and segment-level forgeries...*
>
> | | FLF2V (seg.) | | | TI2V (seg.) | | | MV2V (obj.) | | |
> |---|---|---|---|---|---|---|---|---|---|
> | Method | F1-Det | F1-Loc | RQ | F1-Det | F1-Loc | RQ | F1-Det | F1-Loc | RQ |
> | DeMamba* | 55.2 | 52.6 | - | 63.6 | 62.4 | - | 41.3 | 41.5 | - |
> | MSLoc-PG | 66.1 | 61.8 | - | 74.1 | 73.6 | - | 57.8 | 54.0 | - |
> | Trace*+PG | 67.3 | 69.5 | 3.78 | 72.6 | 72.4 | 3.99 | 63.7 | 68.6 | 3.75 |
> | **MSLoc** | 68.3 | 71.5 | 4.01 | 73.4 | 73.1 | 4.13 | 65.5 | 70.8 | 3.83 |
>
> As expected, object-level editing (MV2V) is hardest due to smaller regions and fewer real↔fake transitions. Per-type results will be added to Tab. 2.
>
> ---
>
> **Q3 (Minor 3):** *...swap vertical positions of Stage I and Stage II in Fig. 5...*
>
> We will update Fig. 5 accordingly.
>
> ---
>
> **Q4 (Major 1):** *...comparative annotation strategy already used in image-level work [1][2]...how to adapt to temporal complexities...*
>
> We agree the motivation is shared, but video adaptation requires: **(1)** vertical concatenation of real/AI videos for frame-wise comparison (multimodal LLMs lack multi-video support); **(2)** boundary context clips for temporal transition reasoning, absent in image-level work; **(3)** structured prompts + manipulation masks for object-level edits. All 28,155 rationales are screened by six inspectors. Details will be expanded in revision.
>
> ---
>
> **Q5 (Major 2):** *...failure case analysis...localization error...localization–rationale consistency...linguistic output error...*
>
> (1) For predictions with IoU ≥ 0.7, mean boundary error is 1.4 s (log-normal distribution, 72.3% in 0.5–2.0 s), showing precise alignment remains hard even with good overall localization.
>
> (2) Rationale Quality (RQ) and sub-dimensions vs. boundary error:
>
> | Bd. Err (s) | RQ | Flu. | Obj. | Type | Detail |
> |---|---|---|---|---|---|
> | 0–0.5 | 4.21 | 4.48 | 4.35 | 4.30 | 3.71 |
> | 0.5–1 | 4.08 | 4.45 | 4.18 | 4.12 | 3.57 |
> | 1–2 | 3.82 | 4.51 | 3.85 | 3.72 | 3.20 |
> | 2–4 | 3.47 | 4.43 | 3.41 | 3.30 | 2.74 |
> | >4 | 3.05 | 4.47 | 2.87 | 2.68 | 2.08 |
>
> RQ drops monotonically (4.21→3.05), confirming localization–rationale correlation.
>
> (3) **Fluency** stays stable (~4.45), while **Object Accuracy** and **Type Accuracy** drop notably (−1.48, −1.62). **Detail Consistency** declines most (3.71→2.08, -1.63), as it compounds object/type errors. Will include in revision.
>
> ---
>
> **Q6 (Major 3):** *...general understanding vs. training-distribution correlation...*
>
> Our benchmark probes this via unseen AIGC tools and unseen video distributions (Tab. 2). On unseen tools, drops are moderate, suggesting no reliance on tool-specific cues. On out-of-domain (OOD) data, degradation is larger. Notably, MSLoc achieves F1-Det=55.0 / F1-Loc=56.3 on OOD vs. 47.2/38.7 for the proposal generator alone, suggesting Stage II captures more transferable cues. Will clarify in revision.
>
> ---
>
> **Q7 (Major 4):** *...DeMamba no longer SOTA [3][4][5]...initial performance...truncation/omission...*
>
> DeMamba supports end-to-end training without task-specific designs, suitable for our benchmark. We reproduced it (DeMamba*, no released weights):
>
> (1) Miss rate: 12.19% (51.06% are <4 s segments), likely due to real–fake transitions within 2 s windows. MSLoc-PG reduces this to 8.73% via boundary-aware optimization.
>
> (2) Truncation rate for segments ≥5 s: both DeMamba* and MSLoc-PG <5.5%, indicating stable predictions suited for proposal generation.
>
> Will cite [3][4][5] in revision.
>
> ---
>
> **Q8 (Major 5):** *...why 10%–20% boundary percentage is optimal...*
>
> | φ | F1-Det | F1-Loc | RQ |
> |---|---|---|---|
> | 5% | 62.4 | 62.0 | 3.72 |
> | 10% | 63.8 | **64.0** | 3.80 |
> | 15% | 63.9 | 63.9 | 3.91 |
> | 20% | **64.0** | 63.8 | 3.99 |
> | 25% | 63.7 | 63.5 | **4.02** |
> | 30% | 63.6 | 63.7 | 4.01 |
> | 40% | 62.8 | 62.5 | 3.82 |
> | 50% | 61.5 | 61.0 | 3.65 |
>
> Performance is stable across φ ∈ [10%, 30%]; φ = 20% is the best trade-off. Too small φ misses boundaries due to proposal imprecision; too large introduces excessive background. Will include in revision.
>
> ---
>
> Thank you again. **Due to space limits, our responses are concise; we warmly welcome further discussion and are happy to elaborate on any point.**

---

> > ### Author Rebuttal · Reviewer_RLFM · 2026-04-03
> >
> > My concerns have been adequately addressed. I am satisfied with the rebuttal and will increase my score accordingly.

---

> > > ### Author Response · Authors · 2026-04-03
> > >
> > > Dear Reviewer RLFM,
> > >
> > > We are delighted to hear that our rebuttal has addressed your concerns and that you have decided to increase your score.
> > >
> > > Your insightful comments and suggestions have been invaluable in strengthening our work. We will ensure that all the discussed points and clarifications are fully integrated into the final version. Furthermore, as promised, our dataset, source code, and project website will be made publicly available to the community.
> > >
> > > Thank you again for your time and for the constructive guidance that helped us improve our paper.
> > >
> > > Best regards,
> > > Authors

---

### Official Review · Reviewer_zLE4 · 2026-03-11

**Soundness:** 4
**Presentation:** 4
**Significance:** 4
**Originality:** 4
**Overall Recommendation:** 5
**Confidence:** 4

**Summary:**

This paper focuses on detecting AI generated content within long videos compared to the usual setup where the detection is done on independent videos for real vs generated content. The authors introduce TASLE a benchmark with more than 10k videos coming from several sources and with three types of video manipulations. In addition to the benchmark, the paper proposed MSLoc a baseline combining a proposal generation module based on DeMamba with a MLLM refinement network to improve the localization and generate an explanation. The experiments show that this baseline outperforms existing methods for detection, localization and explanation metrics.

**Compliance With Llm Reviewing Policy:**

Affirmed.

**Final Justification:**

The rebuttal addressed my questions and reinforced my prior assessment, I am still vouching for acceptance.

**Key Questions For Authors:**

1) Could you provide more information on the human correction for the explanation rationale like giving examples of types of errors or how frequently rationales needed to be corrected?
2) Why not reporting the mAP metric besides the F1_Det?

**Limitations:**

yes

**Strengths And Weaknesses:**

Strengths:
1) This paper defines a new task which is not addressed in the literature. Instead of predicting whether a short clip is real or fake, they aim at localizing AI content in long videos. This is closer to practical applications where AI clips can inserted in a real video. The paper is also very clear about the problems of existing approaches: boundary insensitivity or interference from real content.
2) TASLE is a well designed comprehensive dataset with several data sources and three types of manipulation with both segment level and object level manipulations. Furthermore, the generated explanation rationales were controlled by 6 humans providing extra credibility to the dataset.
3) The proposed baseline is also well designed, makes sense and outperforms existing recent methods such as BusterX++ or DeMamba for detection and Trace for localization. Furthermore, the authors provide thorough ablation studies.

Weaknesses:
1) Besides figures 7 and 8, there is a lack of information regarding the human correction for explanation rationales. We don't know what are the types of errors and how frequent they are.
2) As the dataset relies on the visible artifacts produced by current video generative models, it might become obsolete. It would be interesting to check that a method trained on older models can generalize to new generative models.
3) In table 2, the out of domain performance is much lower than in domain so the forensic aspect is not generalizable and seems to learn datasets specific artifacts.

---

> ### Author Rebuttal · Authors · 2026-03-31
>
> Dear Reviewer zLE4, thanks for your thoughtful review and positive feedback. We address your concerns point-by-point below.
>
> ---
>
> **$\bullet$ Q1 (W1 & Key Question 1):** What types and frequencies of errors occur in human correction of explanation rationales?
>
> Six trained annotators reviewed each rationale using five hierarchical criteria: (1) visual object correctness, (2) manipulation type accuracy, (3) object-description consistency, (4) description clarity, and (5) format or anomaly completeness. Errors were counted hierarchically (i.e., if an earlier category was incorrect, later categories were not additionally counted). Across 28,155 samples, the most frequent issue was incorrect visual object identification (12.3%), including misidentified or omitted key objects. Manipulation type errors accounted for 9.8% (e.g., describing a motion anomaly as a texture issue), followed by object-description inconsistency (2.7%), insufficient clarity (5.4%; e.g., vague descriptions such as "looks unnatural"), and format/truncation issues (1.2%). For example, one original rationale described a "facial" anomaly; the actual artifact was unnatural hair motion, corrected to "hair." These results suggest that model-generated rationales primarily struggle with accurate object identification, motivating human review in our pipeline. We will include this analysis and additional examples in the appendix.
>
> ---
>
> **$\bullet$ Q2 (W2):** Can methods trained on older models generalize to new generators?
>
> | Method | $F1_{Det}$ | $F1_{Loc}$ | RQ |
> |---|---|---|---|
> | D3 | 25.3 | 20.7 | - |
> | DeMamba$^{\ast}$ | 36.5 | 29.8 | - |
> | Trace$^{\ast}$ | 28.4 | 27.6 | 2.94 |
> | Our MSLoc-PG | 43.7 | 35.2 | - |
> | Our MSLoc | **51.2** | **50.8** | **3.61** |
>
> To evaluate generalization to emerging generators, We constructed a zero-shot test set using Seedance 2.0 (Feb. 2026), the latest closed-source FLF2V model not used during training, with 50 manipulated long videos and 50 authentic videos. Short-video detectors like D3 and DeMamba$^{\ast}$ degrade substantially on this new generator, whereas MSLoc-PG retains high proposal recall, suggesting boundary-aware proposal generation generalizes better to unseen generators. Moreover, MSLoc achieves $F1_{Det}$=51.2 and $F1_{Loc}$=50.8, indicating that the temporal, boundary, and physical inconsistency cues remain informative for the latest generators. These results suggest that TASLE is not tied only to outdated artifact patterns. We view this work as an early exploration of this new long-video forensic setting, and hope it will encourage future research on more robust methods and continual adaptation to emerging generators. We will include this analysis and the corresponding data in the revised manuscript.
>
> ---
>
> **$\bullet$ Q3 (W3):** Out-of-domain performance is much lower than in-domain, suggesting the model learns dataset-specific artifacts.
>
> We agree that out-of-domain generalization remains challenging. However, the observed gap does not imply that the model merely learns dataset-specific artifacts. TASLE probes generalization along two dimensions: unseen AIGC tools and unseen video distributions (Tab. 2). On unseen AIGC tools, most methods show only moderate performance degradation, suggesting they capture transferable temporal and artifact cues rather than generator-specific shortcuts. Moreover, performance drops much more substantially on out-of-domain video distributions, indicating a harder distribution shift. This is further reflected by the poor performance of the training-free D3 baseline ($F1_{Det}$: 27.4, $F1_{Loc}$: 22.5), suggesting difficulty beyond simple dataset-specific fitting. Notably, our MLLM-based refinement stage improves robustness: MSLoc achieves $F1_{Det}$=55.0, $F1_{Loc}$=56.3 on out-of-domain data, compared to 47.2/38.7 for MSLoc-PG alone (Tab. 2). This suggests our method partially captures transferable cues, though out-of-domain generalization remains far from solved. We will clarify this discussion in the revised manuscript.
>
> ---
>
> **$\bullet$ Q4 (Key Question 2):** Why not report mAP besides $F1_{Det}$?
>
> We report $F1_{Loc}$ rather than mAP for temporal localization. $F1_{Loc}$, from E.T. Bench (NeurIPS 2024), computes precision and recall over predicted segments at IoU thresholds $\{0.1, 0.3, 0.5, 0.7\}$. We choose it over mAP for two reasons. First, $F1_{Loc}$ penalizes false positives on authentic videos: any predicted segment in a negative video is a false positive, crucial in forensic scenarios where authentic videos are common. In contrast, mAP is designed for ranked predictions rather than explicit false-positive control. Second, mAP requires confidence scores for precision-recall curves, while MLLM-based models output temporal coordinates without reliable scores. $F1_{Loc}$ operates directly on predicted segments, making it better matched to our model's output form.
>
> ---
>
> We appreciate your constructive feedback and welcome any further questions.

---

> > ### Author Rebuttal · Reviewer_zLE4 · 2026-04-03
> >
> > I thank the authors for the rebuttal, I will keep my positive rating.

---

> > > ### Author Response · Authors · 2026-04-03
> > >
> > > Dear Reviewer zLE4,
> > >
> > > We sincerely thank you for your positive feedback and for acknowledging that our rebuttal has addressed your concerns.
> > >
> > > We will incorporate the discussed improvements into the final version of our paper and, as promised, make the dataset, tools, and source code publicly available to the research community.
> > >
> > > Thank you again for your valuable time and constructive guidance.
> > >
> > > Best regards,
> > > Authors

---

### Official Review · Reviewer_vt7d · 2026-03-12

**Soundness:** 3
**Presentation:** 3
**Significance:** 3
**Originality:** 2
**Overall Recommendation:** 4
**Confidence:** 4

**Summary:**

This paper introduces Temporal AI-Generated Segment Localization and Explanation, a new forensic task focused on untrimmed long-form videos. Unlike traditional methods that classify short, independent clips as entirely real or fake, this work addresses the realistic scenario where AI-generated content is sparsely embedded within authentic footage. To support this task, the authors contribute TASLE, a large-scale benchmark of 12,472 videos with over 28,000 rationales covering diverse manipulation patterns (First-Last-Frame-to-Video, Text-Image-to-Video, and Mask-Video-to-Video). They also propose MSLoc, a coarse-to-fine baseline that uses a boundary-sensitive proposal module to scan long videos and an MLLM-based refinement module to provide precise localization and interpretable reasoning.

**Compliance With Llm Reviewing Policy:**

Affirmed.

**Final Justification:**

The authors addressed my concerns. I will keep my score.

**Key Questions For Authors:**

Table 2 shows that while MSLoc-PG improves localization, the overall $F1_{Det}$ and $F1_{Loc}$ still drop significantly on out-of-domain data compared to in-domain "seen" AIGC types. To what extent can the "Difference-Aware Modeling" (DAM) in the second stage recover from imprecise proposals in out-of-domain scenarios, and have authors considered an end-to-end variant that could potentially mitigate the "hard-threshold" loss of the proposal stage?

**Limitations:**

Yes

**Strengths And Weaknesses:**

Strengths:
1. The TASLE dataset is significant in scale and diversity, utilizing 11 different source domains and multiple advanced generative paradigms to ensure high-quality, challenging content.
2. Unlike many "black-box" detectors, this work provides fine-grained, segment-level rationales that describe specific visual anomalies and boundary inconsistencies.
3. The authors evaluate the model against both unseen AIGC types and out-of-domain data, providing a transparent view of current forensic limitations

Weaknesses:
1. The model's effectiveness depends on the visibility of generative artifacts. As AI video generators improve and visual flaws become more subtle, the current feature-based detection may face diminishing returns.
2. As a cascaded framework, the final performance is strictly bounded by the recall of the initial proposal stage; any manipulated segments missed by MSLoc-PG cannot be recovered during the refinement stage.
3. The current benchmark focuses exclusively on visual forensics. While the paper notes the difference from Deepfake detection, it does not yet incorporate audio-visual synchronization or physical logic checks (e.g., gravity, fluid dynamics) as core detection signals.

---

> ### Author Rebuttal · Authors · 2026-03-31
>
> Dear Reviewer vt7d, thanks for your thoughtful review and positive feedback. We address your concerns point-by-point below.
>
> ---
>
> **$\bullet$ Q1 (W1):** Detection effectiveness may diminish as AI video generators improve and visual flaws become more subtle.
>
> **Our work aims to fill this research gap by advancing AIGC forensics from short-video detection to long-video manipulated segment localization and explanation.** We agree that, methods relying solely on low-level appearance artifacts may become less effective. To mitigate this, TASLE reduces reliance on easy visual artifacts: it incorporates recent AIGC tools (Tab. 9, Fig. 4), reference-conditioned segment generation, and object-level MV2V editing, making manipulated content substantially closer to real videos. Moreover, TASLE's cues include temporal and physical inconsistencies (Fig. 3(e)(f)), such as boundary/contact anomalies, incongruous object or limb motion, and motion inertia discontinuities, which are less susceptible to improved visual fidelity. We acknowledge this is an evolving problem, and future work will likely require continuously updating TASLE with stronger generators and moving beyond appearance-based detection toward physics- and motion-aware modeling. We will include this discussion in the final version.
>
> ---
>
> **$\bullet$ Q2 (W2):** As a cascaded framework, performance is strictly bounded by the recall of the proposal stage; missed segments cannot be recovered during refinement.
>
> We agree that this is an inherent limitation of cascaded frameworks. **However, our two-stage design is motivated by the need to jointly achieve efficient scanning and precise localization with explanations:** end-to-end MLLMs suffer from interference from irrelevant real content (e.g., Trace$^{\ast}$, $F1_{Det}$ 42.0 in Tab. 2), while specialized detectors like DeMamba$^{\ast}$ lack explanation capability. To alleviate this, MSLoc-PG uses a boundary-aware four-class formulation, which substantially improves boundary-sensitive proposal quality over binary-classification DeMamba$^{\ast}$ (e.g., +12.6 $F1_{Det}$ and +10.8 $F1_{Loc}$ on in-domain data in Tab. 2). Our first stage is thus a dedicated boundary-sensitive localization module, not a simple coarse filter. We will further clarify this trade-off and its limitation in the final version.
>
>
> ---
>
> **$\bullet$ Q3 (W3):** The current benchmark focuses exclusively on visual forensics. While the paper notes the difference from Deepfake detection, it does not yet incorporate audio-visual synchronization or physical logic checks (e.g., gravity, fluid dynamics).
>
> We intentionally scope this benchmark to visual forensics. Regarding audio-visual synchronization, we agree that it is an important direction, but incorporating it would require jointly addressing long-video AIGC segment localization, realistic audio generation/insertion, and audio-visual multimodal fusion. Moreover, current generators (e.g., Wan2.1, Kling, Vidu, LTXVideo) generally produce silent videos, making consistent audio manipulation itself a specialized challenge (e.g., text-to-speech and audio editing). We therefore focus on visual forensics in this work and view audio as an important future extension. Regarding physical logic checks, TASLE already includes physically related cues (Fig. 3(e)(f)): contact point/boundary anomalies, incongruous limb/object motion, and motion inertia discontinuities, though it does not model full physical simulation. We will clarify this scope and discussion in the final version.
>
> ---
>
> **$\bullet$ Q4 (Key Question):** $F1_{Det}$ and $F1_{Loc}$ drop significantly on out-of-domain data. To what extent can the DAM in the second stage recover from imprecise proposals in out-of-domain scenarios? Have authors considered an end-to-end variant potentially mitigating the "hard-threshold" loss of the proposal stage?
>
> We conducted an ablation experiment on out-of-domain data: without DAM, the backbone achieves $F1_{Det}$=52.7 and $F1_{Loc}$=54.0, while adding DAM improves them to 54.0 (+1.3) and 55.5 (+1.5), confirming its benefit in out-of-domain scenarios.
>
> Regarding end-to-end alternatives, they in principle alleviate the hard-threshold issue of proposal selection. However, they face two major challenges under the current long-video forensic setting: sparse manipulated segments are easily overwhelmed by abundant authentic content under uniform whole-video sampling, and existing temporal grounding MLLMs typically assume that a target event exists, which is mismatched with forensic scenarios containing fully authentic videos. Empirically, end-to-end Trace$^{\ast}$ ($F1_{Det}$=42.0) performs substantially worse than our MSLoc (70.1) in Tab. 2. Our method is intended as a simple yet effective two-stage baseline for this new problem setting, serving as a starting point rather than an endpoint. We hope it can encourage future end-to-end designs.
>
> ---
>
> We appreciate your constructive feedback and welcome any further questions.

---

> > ### Author Rebuttal · Reviewer_vt7d · 2026-04-03
> >
> > Thanks for the rebuttal. I will keep the score.

---

> > > ### Author Response · Authors · 2026-04-03
> > >
> > > Dear Reviewer vt7d,
> > >
> > > We sincerely thank you for your positive acknowledgement and for maintaining your support for our work.
> > >
> > > Your insightful guidance throughout the review process has been instrumental in strengthening this paper. We will ensure that all the discussed points and clarifications are properly integrated into the final version.
> > >
> > > Best regards,
> > > Authors

---

### Official Review · Reviewer_L1j4 · 2026-03-13

**Soundness:** 2
**Presentation:** 2
**Significance:** 2
**Originality:** 2
**Overall Recommendation:** 3
**Confidence:** 4

**Summary:**

The paper introduces a new forensic setting for detecting, localizing, and explaining AI-generated manipulated segments inside untrimmed long videos, arguing that prior video-forensics work focuses too much on short clips that are entirely real or fake and therefore misses the more realistic mixed real–fake setting. To address this, the authors build TASLE, a benchmark of 12,472 long videos with temporal boundaries, authenticity labels, and boundary-/object-level rationales, and propose MSLoc, a coarse-to-fine two-stage baseline that first scans long videos with a boundary-aware proposal generator and then refines candidate segments with an MLLM to obtain precise localization and natural-language explanations. Experiments show that MSLoc outperforms transferred short-video detectors and prior localization baselines on detection, localization, and rationale quality, while also highlighting that boundary modeling is especially important in long-video forensic analysis.

**Compliance With Llm Reviewing Policy:**

Affirmed.

**Key Questions For Authors:**

refer to the weakness

**Limitations:**

refer to the weakness

**Strengths And Weaknesses:**

Weakness:
1. The paper is centered on a new dataset plus a baseline, but the methodological novelty of the model itself feels moderate; MSLoc is mainly a sensible two-stage adaptation of existing detector and grounding-MLLM ideas.
2. The benchmark is still synthetic/manufactured: manipulated segments are inserted using AI generators into real videos, which may not fully capture real-world manually adversarial editing behavior.
3. The rationale annotations are not purely human-authored; they are first generated by Qwen3-VL-235B and then manually screened, so the explanations may inherit biases or stylistic artifacts from the annotation model.
4. The paper compares mostly against transferred short-video detectors and Trace-style localization models, but it does not deeply compare against a broader set of stronger end-to-end long-video baselines or more recent forensic foundation models, so the empirical positioning could be more comprehensive.
Strengths:
1. The paper tackles a genuinely important and timely problem: localized AI manipulation in long videos, which is more realistic than the common short-clip real-vs-fake setup.
2. TASLE is somewhat meaningful with temporal boundaries, mixed real–fake videos, and rich explanation annotations, enabling evaluation beyond binary detection.

---

> ### Author Rebuttal · Authors · 2026-03-31
>
> Dear Reviewer L1j4, thanks for your thoughtful review and positive feedback. We address your concerns point-by-point below.
>
> ---
>
> **$\bullet$ Q1 (W1):** The methodological novelty of MSLoc feels moderate; it mainly adapts existing detector and grounding-MLLM ideas.
>
> MSLoc is a simple yet effective baseline for this new long-video forensic setting, but not a straightforward combination of existing detector and grounding-MLLM components.
>
> Existing grounding MLLMs (e.g., Trace$^{\ast}$, Tab. 2) perform poorly on long videos due to dense sampling ($F1_{Det}$=42.0). Specialized detectors like DeMamba$^{\ast}$ identify manipulated regions but lack precise temporal boundaries or explanations. Our MSLoc bridges this gap by jointly enabling efficient long-video scanning, precise localization, and interpretable rationales.
>
> Importantly, our framework includes task-specific designs: **First**, MSLoc-PG introduces a boundary-aware four-class formulation $\mathcal{Y} = \{real, fake, real \to fake, fake \to real\}$, explicitly modeling transition boundaries and improving proposal quality over DeMamba$^{\ast}$ ($F1_{Loc}$: 64.8 vs. 54.0, Tab. 2). **Second**, DAM extracts inter-frame change/invariance tokens to preserve temporal transition cues. **Third**, $\mathcal{L_{AA}}$ supervises anomaly-aware tokens for manipulation-category prediction, improving explanation quality and localization (MSLoc vs. Trace$^*$ with MSLoc-PG, $F1_{Loc}$: 72.2 vs. 70.9, Tab. 2).
>
> We view MSLoc provides a good starting point for future research. **We will release our dataset, method, website, and benchmarking results for research community.**
>
> ---
>
> **$\bullet$ Q2 (W2):** The benchmark is synthetic/manufactured and may not capture real-world adversarial editing behavior.
>
> TASLE includes realistic adversarial editing beyond inserting AI-generated segments. The *Out-of-Domain Data* split (Tab. 2) is constructed from manually tampered TVSum videos, with annotators controlling manipulated segments, editing strategy, and prompt design, ensuring visual and semantic coherence. This split simulates realistic adversarial behavior and is substantially more challenging.
>
> Beyond segment-level generation (FLF2V/TI2V), TASLE includes object-level manipulation via VACE (MV2V), producing targeted edits by replacing or removing salient objects (e.g., pedestrians, vehicles) under mask-conditioned generation (Sec. 3). These edits more closely resemble realistic adversarial manipulation. TASLE goes beyond purely synthetic composition and captures important characteristics of real-world adversarial editing. We will continue expanding TASLE with richer manually adversarial editing scenarios.
>
> ---
>
> **$\bullet$ Q3 (W3):** Rationales are not purely human-authored; they inherit biases or stylistic artifacts from the annotation model.
>
> During annotation, we explored multiple model families (Qwen, GPT, Gemini) and analyzed accuracy and stylistic tendencies. Our rationales follow a structured format: anomaly objects, anomaly types, and descriptive explanations (Fig. 2, Fig. 3(e)(f)). Stylistic variation is naturally constrained; the main criterion is correctness of anomaly identification.
>
> To encourage diversity, we manually selected high-quality rationales as in-context examples. All generated rationales were manually screened and corrected by six inspectors (Sec. 3). Final annotations are human-verified rationales with controlled style and content.
>
> ---
>
> **$\bullet$ Q4 (W4):** Comparisons could be more comprehensive against stronger end-to-end baselines or recent forensic foundation models.
>
> This is the first exploration of long-video AIGC segment localization with explanations; no directly comparable method covers all task aspects.
> We selected baselines from three closely related directions:
> (1) short-video AIGC detectors (e.g., D3, DeMamba$^{\ast}$), effective at detection but do not provide temporal explanations;
> (2) forensic MLLM-style models (e.g., BusterX++$^{\ast}$, fine-tuned Qwen3-VL$^{\ast}$), generating explanations on short windows at high cost; and
> (3) temporal grounding MLLMs (e.g., Trace$^{\ast}$), supporting localization but easily distracted by authentic content.
> These three groups are representative of the most relevant research directions related to our task and provide a meaningful empirical context for validating MSLoc. We will cite and discuss more recent works in the revision.
>
> ---
>
> **$\bullet$ Q5 & Q6 (Strengths):** **Q5:** The problem of localized AI manipulation in long videos is genuinely important and timely. **Q6:** TASLE is meaningful with temporal boundaries, mixed real–fake videos, and rich explanation annotations.
>
> We sincerely appreciate the reviewer's recognition of TASLE and our proposed benchmark. We will publicly release the dataset and code.
>
> ---
>
> Thanks again for your constructive feedback and welcome any further questions.

---

> > ### Author Rebuttal · Reviewer_L1j4 · 2026-04-06
> >
> > The authors have addressed part of my concerns. However, I still feel that the proposed methodology offers limited substantive novelty, and the manuscript does not yet clearly establish a meaningful knowledge gap for the task. Therefore, I keep my original perspective.

---

> > > ### Author Response · Authors · 2026-04-07
> > >
> > > Dear Reviewer L1j4,
> > >
> > > Thank you for your further comments.  We would like to further clarify the central contribution and knowledge gap addressed by our work.
> > >
> > > Prior AIGC forensics mainly studies short clips that are entirely real or entirely fake. These works are important, but real-world misuse is often different: only a few short manipulated segments are inserted into an otherwise authentic long video. In this setting, a practical system must answer three questions simultaneously: whether manipulation exists, where it occurs, and why it is suspicious. This motivates our task, Temporal AI-Generated Segment Localization and Explanation (TASLE). **To the best of our knowledge, this is the first work on long-video AIGC forensics that jointly requires detection, temporal localization, and explanation.**
> > >
> > > **Existing methods cannot solve this problem well.** Short-video AIGC detectors can identify fake content, but cannot localize manipulated segments or explain them. Grounding MLLMs can localize and explain, but are easily distracted by the large amount of authentic content in long videos and often miss short manipulated segments.
> > >
> > > **MSLoc is designed specifically for this gap.** Its first stage introduces a boundary-aware proposal formulation that explicitly models transitions between authentic and manipulated content, which are ignored by prior binary detectors. The second stage further refines the localization and provides interpretable rationales. **This design is effective.** Compared with the strongest adapted baseline, DeMamba, our proposal stage improves $F1_{Loc}$ from 54.0 to 64.8 and reduces the missed-segment rate from 12.19% to 8.73%, especially for short segments near real/fake transitions.
> > >
> > > Therefore, we respectfully believe that our work is more than “a dataset plus a baseline”: it introduces the first task, benchmark, and dedicated solution for long-video manipulated segment localization and explanation. **We will publicly release the dataset, code, benchmark, and website to support future research.**
> > >
> > > Best regards,
> > > Authors

---

### Decision · Program_Chairs · 2026-04-30

**Decision:**

Accept (regular)

**Comment:**

This paper tackles an important problem, detecting localized manipulations in real, long-duration videos, by introducing a large-scale dataset and a method to solve it. Three of the reviewers argued for acceptance, and one did not. The negative reviewer has concerns about using generative AI to inject localized manipulations in the dataset, since these may not be representative of manually-inserted manipulations. While this may be true, automated manipulations are a very significant concern, much more so than manual ones, and hence this criticism is not shared by the AC. The other reviewers appreciate the significance of the dataset and the proposed method, giving the paper positive ratings.